# Waterlogging Priming Enhances Hypoxia Stress Tolerance of Wheat Offspring Plants by Regulating Root Phenotypic and Physiological Adaption

**DOI:** 10.3390/plants11151969

**Published:** 2022-07-28

**Authors:** Kai Feng, Xiao Wang, Qin Zhou, Tingbo Dai, Weixing Cao, Dong Jiang, Jian Cai

**Affiliations:** National Technique Innovation Center for Regional Wheat Production, Key Laboratory of Crop Physiology and Ecology in Southern China, Ministry of Agriculture, National Engineering and Technology Center for Information Agriculture, Nanjing Agricultural University, Nanjing 210095, China; 2016201011@njau.edu.cn (K.F.); xiaowang@njau.edu.cn (X.W.); qinzhou@njau.edu.cn (Q.Z.); tingbod@njau.edu.cn (T.D.); caow@njau.edu.cn (W.C.); jiangd@njau.edu.cn (D.J.)

**Keywords:** waterlogging priming, root, hypoxia stress tolerance, anaerobic respiration, aerenchyma, wheat

## Abstract

With global climate change, waterlogging stress is becoming more frequent. Waterlogging stress inhibits root growth and physiological metabolism, which ultimately leads to yield loss in wheat. Waterlogging priming has been proven to effectively enhance waterlogging tolerance in wheat. However, it is not known whether waterlogging priming can improve the offspring’s waterlogging resistance. Here, wheat seeds that applied waterlogging priming for one generation, two generations and three generations are separately used to test the hypoxia stress tolerance in wheat, and the physiological mechanisms are evaluated. Results found that progeny of primed plants showed higher plant biomass by enhancing the net photosynthetic rate and antioxidant enzyme activity. Consequently, more sugars are transported to roots, providing a metabolic substrate for anaerobic respiration and producing more ATP to maintain the root growth in the progeny of primed plants compared with non-primed plants. Furthermore, primed plants’ offspring promote ethylene biosynthesis and further induce the formation of a higher rate of aerenchyma in roots. This study provides a theoretical basis for improving the waterlogging tolerance of wheat.

## 1. Introduction

With global climate change, the frequency and duration of extreme rainfall episodes are increasing. Wheat (*Triticum aestivum* L.) is one of the most important crops that provides essential elements for the demand of humans [1]. Waterlogging is one of the limiting factors for wheat production. Globally, 10–15 million ha of wheat are affected by waterlogging every year, resulting in a 15–20% loss of wheat yield [2,3]. The middle and lower reaches of the Yangtze River are one of the main wheat production areas in China. However, wheat production in this area was seriously affected by waterlogging stress, which led to a 20% reduction in yield [4]. Therefore, improving the waterlogging tolerance of wheat could be an effective measure to ensure food security.

The impact of waterlogging on plants results from the low oxygen environment in the roots. The root is the earliest organ to feel hypoxia stress and leads to a series of physiological metabolic changes. The formation of aerenchyma and high root porosity are important adaptive characteristics that contribute to waterlogging tolerance [5]. Under waterlogging stress, the accumulation of ethylene in the root activates programmed cell death of root cortical cells and induces the formation of aerenchyma [6,7,8], which enhances the diffusion of O_2_ to the root, so as to promote energy supply and alleviate hypoxia stress in wheat [9].

The accumulation of carbohydrates contributes to root fermentation during anaerobic stress and enhances tolerance to anaerobic stress in wheat. In this process, sucrose synthase (Susy) catalyzes sucrose hydrolysis and plays a key role in providing a sufficient substrate for anaerobic respiration during hypoxia stress [10]. Under hypoxia stress, the expression of the ADH gene, which encodes alcohol dehydrogenase (ADH), increases, inducing ethanol fermentation to maintain a high energy level in roots [11]. Compared with the sensitive genotype, the root of the tolerance genotype wheat could accumulate more sugar to promote anaerobic respiration and produce the energy needed for nutrient absorption [12,13,14]. The energy produced by glycolysis and ethanol fermentation can temporarily alleviate the energy shortage caused by a lack of oxygen, which is of great significance for plants to maintain growth under hypoxia stress. In addition, when plants are subjected to waterlogging stress, they produce excessive reactive oxygen species and cause oxidative damage to cell tissues [15]. The roots of wheat seedlings can respond to the harmful effects of hypoxia stress through their antioxidant system, including the increased activities of antioxidant enzymes SOD, POD and CAT [16], and the enhanced ascorbic acid glutathione cycle (AsA-GSH) under waterlogging, especially in primed wheat [17]. They jointly force to reduce the destructive effect of ROS and improve the tolerance to hypoxia stress.

It has been found that abiotic stress priming can increase their tolerance to subsequent stress in crop plants, and the tolerance can be transferred to the next generation [18,19]. The previous study found that drought priming in the parent’s generation induced the tolerance of offspring to drought stress that occurred at the filling stage in wheat [20]. Similarly, heat priming in the parent’s generation can induce the tolerance of offspring to high-temperature stress in wheat [21]. However, the effect of waterlogging priming on the stress tolerance of offspring wheat plants was not known. Therefore, we hypothesized that parents’ waterlogging priming could improve the hypoxia tolerance of offspring wheat plants. In the current study, the morphological response to hypoxia stress of wheat offspring plants was observed. Glycolysis, anaerobic respiration and antioxidant system in root were evaluated to verify whether the hypoxia tolerance can be enhanced compared to non-primed plants.

## 2. Results

### 2.1. Effect of Waterlogging Primed Seedlings on Plant Biomass in Wheat Offspring Plants under Hypoxia Stress

There was no significant difference among T0C, T1C, T2C, T3C in fresh weight and dry weight of shoot and root. Under hypoxia stress, T1W, T2W, T3W showed significantly higher shoot fresh weight, root fresh weight and root dry weight than T0W. No significant differences were found among T0W, T1W, T2W, T3W in the dry weight of the shoot (Figure 1).

### 2.2. Effect of Waterlogging Primed Seedlings on Leaf Photosynthesis and Chlorophyll Fluorescence in Wheat Offspring Plants under Hypoxia Stress

There were no significant differences in the net photosynthetic rates (Pn), stomatal conductance (gs), the maximum quantum efficiency of photosystem Ⅱ (Fv/Fm) and actual photochemical efficiency (ΦPSII) between the different primed treatments of control in offspring seedlings (Figure 2). However, under hypoxia stress, compared with the control, the Pn of T1W, T2W and T3W decreased by 15.8%, 13.3% and 13.7%, respectively, which was much lower than the 28.4% of T0W. Meanwhile, compared with non-primed plant offspring (T0W), the primed wheat offspring (T1W, T2W, T3W) showed higher Fv/Fm (14.85%, 20.67% and 21.31%, respectively) and ΦPSII (9.0%, 8.29% and 8.29%, respectively). The gs of T1W, T2W, T3W had a trend higher than that of T0W, but there were no significant differences in these parameters among T1W, T2W and T3W.

### 2.3. Effect of Waterlogging Primed Seedlings on ROS Scavenging Capacity in Wheat Offspring Roots under Hypoxia Stress

Reactive oxygen species were over-produced in wheat roots under hypoxia stress (Table 1). Under hypoxia stress, compared with the control, the O_2_^−^ production rate of T1W, T2W and T3W was increased by 6.90%, 4.20% and 9.29%, respectively, much lower than the 18.01% of T0W. Meanwhile, the H_2_O_2_ content of T1W, T2W and T3W increased by 44.57%, 35.22% and 37.97%, respectively, much lower than the 73.71% of T0W. As a result, the content of MDA increased significantly under hypoxia stress, among which T0W increased the most (Table 1), and the primed wheat offspring (T1W, T2W, T3W) showed a lower content of MDA (12.28%, 15.08% and 21.40%, respectively) than the non-primed wheat offspring (T0W), significantly.

Compared with control, the SOD activity of T1W, T2W, T3W decreased by 6.90% 4.20% 9.29%, which decrease was still lower than T0W (18.01%). Meanwhile, the primed wheat offspring plants (T1W, T2W, T3W) showed lower activity of CAT (7.57%, 10.76%, 10.16%) and APX (1.15%, 10.98%, 9.36%) than the non-primed wheat offspring plant (T0W). Among them, the APX activity of T1W did not show any significant difference. In addition, the activities of GR, MDHAR and DHAR increased significantly under hypoxia. The primed wheat offspring plants (T1W, T2W, T3W) showed higher activity of GR (31.98%, 39.38%, 26.68%), MDHAR (8.86%, 7.14%, 13.71%) and DHAR (11.60%, 14.20%, 13.00%) than the non-primed wheat offspring plant (T0W).

Moreover, the contents of antioxidant ASA and GSH increased under hypoxia stress. The contents of ASA and GSH in priming treatment (T1W, T2W and T3W) were 51.85%, 51.80%, 48.15% and 19.58%, 20.30%, 21.67% higher than those in T0W, respectively. Meanwhile, the total content of AsA + DHA and GSH + GSSG increased significantly under hypoxia stress. Priming treatments were 30.12%, 32.53%, 36.14% and 14.47%, 15.37%, 16.95% higher than T0W. Thus, the priming treatment has a high ratio of ASA/DHA and GSH/GSSG, which were 26.53%, 20.41%, 10.2%, and 41.72%, 39.53%, 37.33% higher than those in T0W, respectively. In addition, there was no significant difference in ROS content and ROS scavenging system between priming treatments (T1W, T2W, T3W) and between no-priming treatments T0C, T1C, T2C and T3C.

### 2.4. Effect of Waterlogging Primed Seedlings on Content of Total Soluble Sugar and Sucrose in Wheat Offspring Plants under Hypoxia Stress

There was no significant difference among T0C, T1C, T2C, T3C in the contents of total soluble sugar and sucrose of leaf and root (Figure 3). Compared with no-primed wheat offspring plants (T0W), the primed wheat offspring plants (T1W, T2W, T3W) showed a higher content of total soluble sugar (11.60%, 10.02% and 15.09%, respectively) and sucrose (21.48%, 25.06% and 23.37%, respectively) in the leaf. And T1W, T2W, T3W also showed a higher total soluble sugar (48.70%, 35.83% and 46.71%, respectively) and sucrose (12.44%, 15.83% and 20.35%, respectively) than T0W in the root. There was no significant difference between primed treatments (T1W, T2W, T3W) under hypoxia stress.

### 2.5. Effect of Waterlogging Primed Seedlings on Fermentation Metabolism in Wheat Offspring Plants under Hypoxia Stress

Under hypoxia stress, the activities of acid invertase and alkaline invertase in the root were decreased and the activity of sucrose synthase (Susy) was increased (Figure 4). The primed offspring plants (T1W, T2W and T3W) were lower than the non-primed offspring plants (T0W) in acid invertase activity (16.11%, 13.48% and 14.16%, respectively) and alkaline invertase activity (6.45%, 8.64% and 6.47%, respectively). And the sucrose synthase activity of T1W (7.44%), T2W (9.24%) and T3W (11.83%) were higher than T0W.

PDC and ADH are key enzymes for anaerobic respiration. Hypoxia stress induced increased PDC and ADH activity, while the priming treatments (T1W, T2W, T3W) further enhanced the activity of PDC and ADH in roots under hypoxia stress (Figure 5). T1W, T2W and T3W increased by 43.1%, 51.9%, 40.9%, and 75.3%, 86.1%, 80.6%, respectively, compared with T0W. The PDC and ADH gene expressions were consistent with enzyme activity (Figure 6). Meanwhile, the ATP content of the root decreased significantly under hypoxia stress. As compared with control, the ATP content of T1W, T2W, T3W was decreased by 9.44%, 6.40%, 11.72%, which were lower than T0W of that 17.10% under hypoxia stress. In addition, there was no significant difference between T1W, T2W and T3W.

### 2.6. Effect of Waterlogging Primed Seedlings on Aerenchyma Formation in Secondary Root in Wheat Offspring Plants under Hypoxia Stress

To assess the effect of hypoxia stress on the formation of aerenchyma, a four-leaf-stage wheat seedling was treated with hypoxia stress, and the formation of aerenchyma at different positions in the secondary root was dynamically observed by the slice. Aerenchyma was observed at a position 40 mm from the root tip until 24 h after hypoxia stress, and with the extension of hypoxia stress, the aerenchyma extended to both ends (Appendix A). According to the results in the Appendix A, only the position of 40 mm from the root tip was selected for the aerenchyma section and observation in priming treatments. Under control, T0, T1, T2 and T3 roots with different priming treatments had no aerenchyma cavity (Figure 7). The aerenchyma formation rate of primed wheat offspring plants (T1W, T2W and T3W) was significantly higher than that of no-primed wheat offspring plants (T0W) in 3 days (28.61%, 24.88% and 26.07%, respectively) and 7days (42.03%, 36.31% and 41.63%, respectively) of hypoxia stress, while there was no significant difference among T1W, T2W, T3W.

### 2.7. Effect of Waterlogging Primed Seedlings on Ethylene Content in Offspring Roots under Hypoxia Stress

Under hypoxia stress, the ethylene release rate in wheat roots was increased (Figure 8). The ethylene release rate of T1W (67.80%), T2W (80.78%) and T3W (71.43%) increased was higher than that of T0W (55.70%). Further, the relative expression of the *TaACS2* gene was measured, the key enzyme of ethylene synthesis. Under hypoxia stress, the expression of the *TaACS2* gene increased significantly, and the plants after waterlogging priming (T1W, T2W, T3W) were higher than those without waterlogging priming (T0W). There was no significant difference in the relative expression of *TaACS2* among T1W, T2W and T3W.

## 3. Discussion

### 3.1. Parental Waterlogging Priming Improved the Tolerance to Hypoxia Stress in Wheat Offspring Plants

Plant biomass accumulation is one of the most intuitive indicators reflecting stress tolerance [22]. Under waterlogging, the seminal root dry mass declines markedly, whereas new adventitious roots develop [23]. The adventitious root growth does not fully compensate for the loss of seminal root DM, and so root DM is reduced [24]. Waterlogging generally reduces the shoot growth of wheat, which results from less tillering, reduced rates of leaf growth and smaller leaf size. As root growth is inhibited more than shoot growth, waterlogging reduces the root as follows: shoot ratio of wheat [2]. Hypoxia pretreatment can improve the hypoxia tolerance of rice roots under late hypoxia stress, but when the pretreatment time exceeds 12 h, the hypoxia tolerance of rice roots does not further increase [25]. Here, parental waterlogging primed plants showed higher biomass accumulation, compared with non-primed offspring plants. The results indicated that the parental waterlogging priming improved the offspring plants’ hypoxia stress tolerance in wheat seedlings.

Short-term effects of waterlogging on Pn could be due to ‘physiological drought’ in plants, which can even wilt, and gs would decrease to conserve water, also resulting in decreased Pn [26]. During long-term waterlogging, factors such as decreases in chlorophyll or other components of the photosynthetic apparatus as a result of negative feedback from carbohydrate accumulation would be likely causes of reduced Pn. The maximum photochemical efficiency of PSII (Fv/Fm) and actual photochemical efficiency (ΦPSII) can reflect the damage status of the PSII reaction center, which is a key indicator of the photosynthetic capacity of the plant [27]. Waterlogging could destroy chloroplast structure and continuously inhibit photosynthetic electron transfer and PSII activity [28]. In this study, the ΦPSII showed the same tendency as the photosynthesis rate, being higher in parental primed seeds compared with non-primed plants under hypoxia stress. These results indicated that parental waterlogging priming improved the offspring plants’ hypoxia stress tolerance by protecting the PSII reaction center and maintaining plant growth in wheat seedlings. 

### 3.2. Parental Waterlogging Priming Reduce the Oxidative Damage in Wheat Offspring Roots under Hypoxia Stress

Waterlogging stress leads to oxidative damage to plants [29,30]. Excess accumulation of ROS, such as superoxide radial (O_2_^−^) and hydrogen peroxide (H_2_O_2_), could lead to cell membrane damage. Malondialdehyde (MDA) content was often used to measure the degree of cell membrane damage [31]. Here, O_2_^−^ release rate, H_2_O_2_ content, and MDA content increased significantly under hypoxia stress, while the primed plants in the parent’s generation were significantly lower than those of non-primed offspring plant roots. The results suggested that parental waterlogging priming could lead to less cell membrane damage under hypoxia stress. 

Plants have formed a series of ROS scavenging systems, including antioxidant enzymes and antioxidants, to remove excess reactive oxygen species. SOD can reduce O_2_^−^ by decomposing it into O_2_ and H_2_O_2_, which is the first barrier against oxidative damage under stress [32]. Waterlogging treatment could increase antioxidant enzyme activity. Tolerant varieties generally have higher antioxidant enzyme activity than sensitive varieties in corn and barley [16,33,34]. ASA and GSH have strong reducibility and can react directly with reactive oxygen species, which are also important ways to scavenge reactive oxygen species in plants. APX, MDHAR, DHAR and GR are the key enzymes in the AsA-GSH cycle, which can catalyze the cyclic regeneration of ASA and GSH [35]. The changes in antioxidant content and redox state (ASA/DHA, GSH/GSSG) can reflect the ability of plants to resist oxidative stress [36]. Here, the antioxidant enzyme activity and antioxidant content of the primed offspring plants were significantly higher than those of non-primed offspring plants in roots. Those results showed that parental primed plants enhanced the ROS of the scavenging capacity in the root, leading to less oxidative damage of wheat seedling roots caused by hypoxia stress, so that primed wheat seedling roots can grow better.

The formation of aerenchyma in the root is one of the important adaptive characteristics of plants under hypoxia stress [37]. Programmed cell death and degradation occur in the cortical cells of plant roots under hypoxia, producing tissue cavities and leading to aerenchyma formation. Aerenchyma not only transports O_2_ from non-waterlogged tissue to the root system but also discharges CO_2_ and toxic volatile substances from waterlogged tissue, which is vital for maintaining the normal physiological metabolism in the cells of waterlogged roots [6,38]. Under hypoxia, wheat plants can regulate the formation of aerenchyma in wheat roots through an ethylene-mediated ROS signaling pathway. Spraying an ethylene precursor in advance can further promote the development and formation of aerenchyma in wheat seedling roots under an anoxic environment [39]. Compared with sensitive barley, the lateral root of waterlogging tolerant barley showed significantly higher porosity and faster formation of aerenchyma [40]. In this study, a position 40 mm from the root tip was selected to observe the aerenchyma under hypoxia stress because it was the earliest and most obvious position for the formation of aerenchyma. The formation rate of root aerenchyma in waterlogging primed offspring plants was significantly higher than that of non-primed offspring plants. Moreover, the ethylene release rate and the expression of the aminocyclopropane carboxylate synthase gene (*TaACS2),* a key gene of the ethylene synthesis pathway, showed the trend of primed treatment > non-primed treatment > normal treatment. The above results showed that waterlogging priming could advance ethylene biosynthesis by inducing *TaACS2* gene expression, which leads to the formation of aerenchyma in wheat roots under hypoxia stress, reducing the adverse effects of hypoxia stress on wheat seedling roots.

### 3.3. Parental Waterlogging Priming Promoted the Energy Metabolism to Hypoxia Stress in Wheat Roots Offspring Plants

The growth of the plant was inhibited under waterlogging or hypoxia stress, but the photosynthesis products could still accumulate, resulting in an increase in soluble sugar content in the leaf [41]. On the one hand, it can be used as an important osmotic regulator to protect the leaf. On the other hand, soluble sugar is transported to the root to provide substrate for glycolysis and anaerobic respiration and alleviate ATP deficiency caused by waterlogging stress [2]. It has been shown that the regulation of glycolysis and fermentation pathways and the accumulation of carbohydrates in roots during hypoxia stress play important roles in prolonging the survival time of plants under hypoxia [42]. In order to ensure the survival or growth of plants, more carbohydrates need to be consumed to produce ATP by anaerobic respiration under hypoxia stress. Carbohydrate starvation has been shown to be one of the causes of hypoxia-induced injury [43]. Sucrose is degraded in plant cells in the following two different ways: the sucrose synthase (Susy) pathway and the invertase (INV) pathway. It was suggested that more carbohydrates be accumulated in primed plants to enhance root growth. Under hypoxia stress, sucrose synthase is the preferred method for sucrose hydrolysis because it retains part of the energy of the glucose–fructose bond [44]. The importance of Susy in hypoxia tolerance has been confirmed by the double mutant plant (*sus1* × *sus4*) that showed lower tolerance to waterlogging [45]. In this study, compared with control, the sucrose content and sucrose synthase activity in roots increased while invertase activity decreased significantly under hypoxia stress, whereas this trend in parental primed plants was more significant than that of non-primed plants. The above results showed that waterlogging priming could promote the supply of soluble sugar to the roots of offspring plants, provide substrates for glycolysis and anaerobic respiration, and maintain ATP synthesis in a short time to respond to the needs of plants.

Under hypoxia stress, aerobic respiration is inhibited, respiratory rate is decreased and plants need to maintain survival by using part of the energy generated by anaerobic respiration pathways. Pyruvate decarboxylase (PDC) and alcohol dehydrogenase (ADH) activity increased in wheat under anaerobic conditions, resulting in more ethanol but also producing more energy to cope with anaerobic stress [14]. Overexpression of *PDC1* and *PDC2* in Arabidopsis has been shown to improve ethanol production and survival under hypoxia conditions [46]. And the *ADH* deletion mutant showed lower survival under hypoxia in Arabidopsis [47,48]. Here, the PDC and ADH enzyme activities and related gene expression in the roots of primed offspring plants were significantly higher than those of non-primed offspring plants. The above results showed that parental waterlogging priming can further promote the anaerobic respiratory response of offspring plants under hypoxia stress, so as to produce more ATP than non-primed offspring plants and alleviate the energy shortage caused by hypoxia stress. This helps maintain the physiological metabolic process of wheat roots, thus alleviating the inhibitory effect of hypoxia stress on root growth.

It is generally believed that the inheritance of stress memory to plant offspring is related to epigenetic modification. For example, continuous drought priming of multiple generations can improve the drought tolerance of offspring in *Polygonaceae* and show a cumulative effect [22]. Salt priming needs two generations to improve the tolerance in *Arabidopsis*. The generation and elimination of these priming effects are closely related to epigenetics [23]. Heat priming and drought priming in the parent’s generation can induce the tolerance of offspring to stress in wheat [20,21]. Different priming treatments will show different effects, which illustrates the mechanism of epigenetic modification is complex. Therefore, we only describe the phenomena and physiological mechanisms of priming to improve the offspring’s hypoxia tolerance but do not explain the lack of cumulative effect of the successive priming on the offspring from the perspective of epigenetic modification. In this study, there was no significant difference in growth status and physiological indexes among different priming treatments (T0C, T1C, T2C and T3C) under control. Under hypoxia stress, there was no significant difference among the offspring wheat seedlings treated by waterlogging priming for one generation (T1), two generations (T2) and three generations (T3), indicating that the waterlogging tolerance of the offspring wheat plants did not increase significantly with the increase in priming generations, which was consistent with the trend of drought priming in wheat [20]. The results showed that parental waterlogging priming could significantly enhance the waterlogging tolerance in wheat offspring plants and had no effect on the growth, development and physiological processes of the wheat offspring under normal growth conditions. Therefore, it could be used as a feasible way to improve the waterlogging tolerance in offspring wheat.

## 4. Materials and Methods

### 4.1. Experimental Design

The experiment was conducted at the experimental station of Nanjing Agricultural University, Jiangsu Province, P. R. China. Ningmai 13 (*Triticum aestivum* L.), which has been largely used in local wheat production, was used to produce the material for this study. Before priming, wheat plants were planted in the pot, and the soil’s relative water content was controlled at 70–80%. The waterlogging priming was applied on the tenth day after anthesis, with half of the pots maintaining a 1–2 cm water layer above the soil for 7 days and the other half were treated by maintaining the relative water content of the soil at 70–80% as a control. After priming, the pots were removed excess water by opening drainage holes and keeping the soil’s relative water content similar to the level of control pots and maintained till harvesting. One generation waterlogging primed seeds (T1), two successive generations waterlogging primed seeds (T2), three successive generations waterlogging primed seeds (T3) were produced as the material for this study (Appendix A).

Here, the harvested grains (T0, T1, T2, T3) were germinated on a mesh float with deionized water until 2 leaf-stage and then transferred to 34-L container (60 plants per container, 22 cm height × 46.5 cm length × 34 cm width) with Hoagland’s solution for 2 weeks. Wheat seedlings were grown in the artificial climate room under 14 h of light, 20/18 °C. At 4-leaf-stage, half of the T0, T1, T2, T3 seedlings were subjected to hypoxia stress through the nutrient solution with N_2_ continuous washing treatment for 7 days, leading to a low oxygen environment in root. The dissolved oxygen content remains at about 1.0 mg/l, which monitored by dissolved oxygen meter (JPSJ-605L, INESA scientific instrument Co. Ltd., Shanghai, China). The other half of the seedlings were kept in nutrient solution with O_2_ continuous washing while maintaining oxygen content at 10 mg/L above (Appendix A). Taken together, eight treatments were established as T0W, T0C; T1W, T1C; T2W, T2C; T3W, T3C.

### 4.2. Photosynthesis and Chlorophyll Fluorescence Parameters

The last expanded leaf was used to measure the gas exchange parameters by the Li-6400 system (LI-COR Inc., Lincoln, NE, USA), with photosynthetically active radiation (PAR) at 800 µmol m^−2^ S^−1^ and the CO_2_ concentration at 380 µmol^−1^.

The chlorophyll fluorescence characteristics were measured using the same leaf by fluorescence imager (CF Imager, Technologia Ltd., Colchester, Essex, UK). The maximum quantum efficiency of photosystem II (Fv/Fm) and the actual photochemical efficiency (ΦPSII) were analyzed using the fluorimager software (Fluorimager 2.2, Technologia Ltd., Colchester, Essex, UK).

### 4.3. Contents of Malondialdehyde (MDA) and Hydrogen Peroxide (H_2_O_2_), Superoxide Anion (O_2_^−^) Release Rate

The extraction of hydrogen peroxide (H_2_O_2_), superoxide anion (O_2_^−^) and malondialdehyde (MDA) were carried out as described by Wang et al. [20]. Weigh 0.5 g leaf or 1.0 g root fresh sample, grind the fresh sample into homogenate on ice with extraction solution, the supernatant was extracted by centrifugation for the determination of above parameters and antioxidant enzyme activity.

The H_2_O_2_ content was measured by hydrogen peroxide detection kit (Beyotime Institute of Biotechnology, Shanghai, China).

The O_2_^−^ release rate was assayed by hydroxylamine method [45]. After mixed reaction system, the mixture was kept at 25 °C for 20 min and the OD value was measured under OD_530_ nm of spectrophotometer.

The MDA content was determined according to Tan et al. [49]. In total, 4 mL TCA-TBA mixture solution was added with 2 mL extraction, boiling water bath for 20 min, centrifugation at 4000× *g* for 10 min, supernatant was taken and read at OD_450_ nm, OD_532_ nm and OD_600_ nm under a visible light spectrophotometer.

### 4.4. Antioxidant Enzyme Activities and Antioxidants Content

The activities of superoxide dismutase (SOD) were measured according to the method of Tan et al. [49]. The reaction mixture was incubated under light and dark to monitor the photoreduction of nitroblue tetrazolium (NBT) at 560 nm and the activity of SOD was expressed as U mg^−1^ protein.

For assaying activity of catalase (CAT), Tan et al. [49]. Method was adopted, which was analyzed by determining the H_2_O_2_ consumption. The reaction mixture was implemented using a spectrophotometer to read the absorbance at 240 nm. CAT activity was calculated according to the change of OD_240_ per minute.

Ascorbate peroxidase activity (APX) was estimated following the technique published by Nakano et al. [50]. The activity was measured by recording the optical density using a spectrophotometer at 290 nm.

Monodehydroascorbate reductase (MDHAR) and glutathione reductase (GR) were analyzed according to Miyake et al. [50]. The activity of those antioxidants was calculated after monitoring the NADPH’s oxidation for 60 s read at 340 nm using a spectrophotometer, and it was expressed as OD_340_ min^−1^ mg^−1^ protein.

Dehydroascorbate reductase (DHAR) was analyzed according to Miyake et al. [50]. The activity was measured by recording the absorption at 265 nm using a spectrophotometer, which determines the DHA reduction rate.

The contents of reduced ascorbate (AsA), dehydroascorbate (DHA), reduced glutathione (GSH) and oxidized glutathione (GSSG) were determined by the method of Gossett et al. [51]. Weigh 0.5 g of fresh root sample, grind it into homogenate in trichloroacetic acid. After low-temperature centrifugation, the supernatant was collected to determine ASA, DHA, GSH GSSG. 

Quickly after the preparation of corresponding reaction mixture, the OD of the reaction mixture was read at 530 nm to calculate AsA and DHA content, the OD of the reaction mixture was read at 412 nm to calculate GSH and GSSG content, and the quantification was conducted using the standard curve prepared from standard solutions.

### 4.5. Sugars Content, Sucrose Hydrolase Enzymes Activity

Crush the dried root and leaf samples and weigh 0.1 g powder to extract three times with 80 % (*v/v*) ethanol. After centrifuging, the supernatant was transferred to a glass tube and the final volume was made up to 20 mL with 80 % ethanol. Total soluble sugar content was measured using the sulfuric acid-anthrone method, sucrose content was determined by resorcin method [52]. 

The determination of acid invertase, alkaline invertase and sucrose synthase (Susy) follows the method of Zrenner et al. [53]. After extracting the enzyme solution from fresh samples by cryogenic centrifuge. Determine the acid invertase and alkaline invertase by controlling the pH of buffer solution in different reaction systems, and the enzyme activity is calculated by the variable of glucose. 

Sucrose synthase (Susy) can catalyze sucrose and UDP to produce fructose and UDPG, and the amount of glucose produced in the reaction system was determined by spectrophotometry with 3,5-Dinitrosalicylic acid method to calculate the enzyme activity.

### 4.6. Activities of Anaerobic Respiration Enzymes, ATP Content

The activities of pyruvate decarboxylase (PDC) and alcohol dehydrogenase (ADH) were measured according to Waters et al. [54]. Half-gram granted wheat root samples were mixed with pre-cooled Tris-HCl extraction. The mixture was centrifuged at 12,000× *g* for 20 min at 4 °C, and the supernatant was used for further analysis. PDC catalyzes pyruvic acid to produce acetaldehyde, and ADH catalyzes the reaction of NADH and acetaldehyde to produce ethanol and NAD+, And then measured the absorbance value at OD_340_, absorption peak of NADH, to calculate the activity of PDC and ADH.

Adenosine 5′-triphosphate (ATP) content was measured according to Liu and Chen et al. [55,56]. In total, 0.5 g root fresh sample was homogenized in 5.0 mL boiling water. The homogenate was incubated in boiling water for 30 min, followed by centrifugation to obtain the supernatant. The extraction solution measures the ATP content according to the user manual of ATP detection kit (Jincheng Bioengineering Institute, Nanjing, China), and measured the absorbance value at OD_636_ to calculate the ATP content.

### 4.7. Root Anatomical Observation

Slice observation of aerenchyma was measured by modifying the description Yamauchi et al. [39]. Store the secondary roots treated with hypoxia for 4 h, 8 h, 12 h, 24 h, 36 h, 72 h and 168 h of each treatment in FAA fixed solution. When slicing, take root segments 10 mm, 20 mm, 30 mm, 40 mm, 50 mm, 60 mm, 70 mm and 80 mm away from the root tip, embed them in 5% agar, slice them with a vibrating slicer (Leica vt1200, Leica Microsystems Ltd, Wetzlar, Germany) with a thickness of 100 µm, observe and take photos with a 10x objective lens of inverted fluorescence microscope (Olympus IX71, Olympus, Tokyo, Japan), and count the formation rate of aerenchyma through the slicing photos (Definition: the ratio of the area formed by aerenchyma to the whole cross-sectional area) was counted.

### 4.8. Ethylene Content

Ethylene was measured by modifying the method of Hattori et al. [57]. The aboveground part was cut from the wheat seedlings, and the remaining underground part was placed in a container containing saturated NaCl solution. The gas in the container is degassed by vacuum pump, and the gas released by wheat seedlings is collected in the test tube by funnel. The collected gas was transferred to a gas chromatographic vial with a rubber stopper and inverted in a saturated NaCl solution. The vial was then righted, a small portion of headspace gas was taken out of the vial with a syringe, and the ethylene content was determined by gas chromatography (GC—17A, Shimadzu Corporation, Kyoto, Japan).

### 4.9. RNA Extraction and Quantitative Real-Time PCR

The root sample was weighed from 50 to 100 mg, ground in liquid nitrogen and placed in 2 mL tube. Total RNA was extracted using RNAiso Plus reagent (Takara Bio, Kusatsu, Shiga, Japan). Then using NanoDrop 2000 (Thermo Scientific, Waltham, MA, USA) for RNA quality determination and concentration determination. Further, according to the concentration of extracted RNA, the cDNA synthesis kit produced by TAKARA was used for reverse transcription. Quantification was performed according to the instructions using the SYBR Premix Ex Taq II kit manufactured by TAKARA. Quantitative real-time PCR was performed on a CFX Connect real-time PCR detection system (Bio-Rad, Hercules, CA, USA) using ChamQ SYBR qPCR Master Mix (Vazyme Bio, Nanjing, China).

Cycle parameters: 95 °C 30 s; 40 cycles of 95 °C for 10 s, 60 °C for 30 s. The melting curve was performed after the PCR cycle. The relative expression level of the gene was calculated according to the 2^−ΔΔCt^ method using the *Actin* gene as a reference gene. The primer design and the reference were listed in Appendix A [58]. Three biological replicates and three technical replicates were performed.

### 4.10. Statistic Analysis

One-way ANOVA (Sigma plot 11.0, Systat Software Inc., San Jose, CA, USA) was applied to analyze the difference between treatments. Significant differences were identified at 0.05 probability level by Duncan’s Multiple Range Test.

## 5. Conclusions

This study illustrated that the progeny of waterlogging primed plants showed higher tolerance to hypoxia stress and that the priming effect is not a simple cumulative effect, so we focused more on the physiological mechanism of priming to improve hypoxia tolerance. Under hypoxia stress, primed plants maintained greater photosynthetic capacity, effective carbohydrate supply from leaf to root, promoted glycolysis and anaerobic respiration and alleviated the lack of energy in the root. Meanwhile, the progeny of primed plants promoted the synthesis of ethylene in the roots, which induced the rapid formation of aerenchyma and triggered more effective antioxidant systems. All of these contribute to enhanced hypoxia stress tolerance (Figure 9).

## Figures and Tables

**Figure 1 plants-11-01969-f001:**
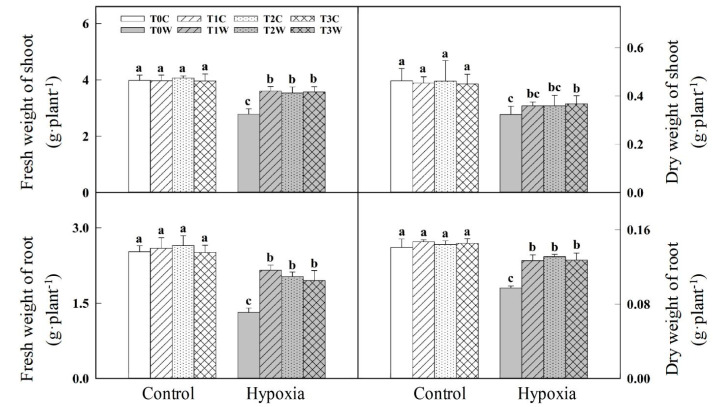
Effects of waterlogging priming on plant biomass accumulation of offspring plants under hypoxia stress in wheat seedling. Note: T0C, no priming + no offspring hypoxia stress; T0W, no priming + offspring hypoxia stress; T1C, one-generation priming + no offspring hypoxia stress; T1W, one-generation priming + offspring hypoxia stress; T2C, two-generation priming + no offspring hypoxia stress; T2W, two-generation priming + offspring hypoxia stress; T3C, three-generation priming + no offspring hypoxia stress; T3W, three-generation priming + offspring hypoxia stress. Data are means ± SE (n = 3). Different lowercase letters indicate the significant difference at *p* < 0.05 level.

**Figure 2 plants-11-01969-f002:**
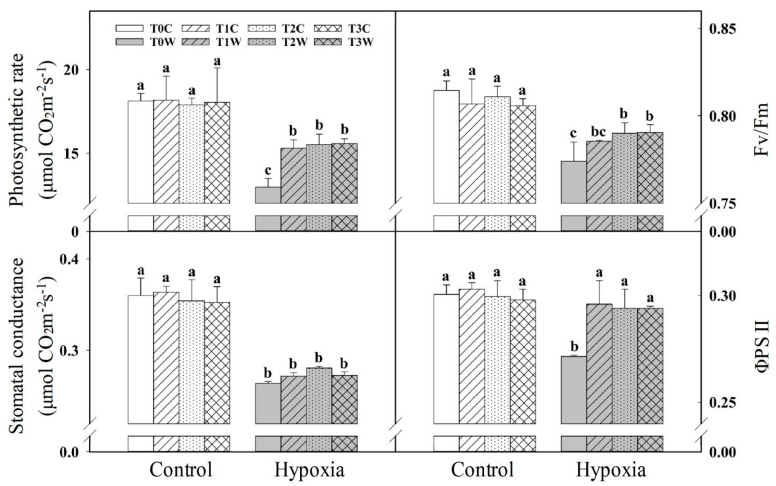
Effects of waterlogging priming on photosynthetic rate (Pn), stomatal conductance (gs), the maximum quantum efficiency of photosystem II (Fv/Fm), and the actual photochemical efficiency (ΦPSII) of offspring plants under hypoxia stress in wheat seedling. Note: T0C, no priming + no offspring hypoxia stress; T0W, no priming + offspring hypoxia stress; T1C, one-generation priming + no offspring hypoxia stress; T1W, one-generation priming + offspring hypoxia stress; T2C, two-generation priming + no offspring hypoxia stress; T2W, two-generation priming + offspring hypoxia stress; T3C, three-generation priming + no offspring hypoxia stress; T3W, three-generation priming + offspring hypoxia stress. Data are means ± SE (n = 3). Different lowercase letters indicate the significant difference at *p* < 0.05 level.

**Figure 3 plants-11-01969-f003:**
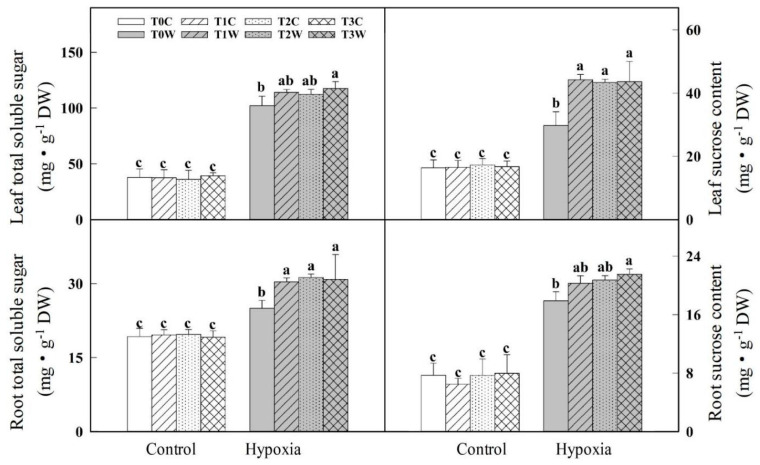
Effects of waterlogging priming on total soluble sugar and sucrose content of offspring plants under hypoxia stress in wheat seedling. Note: T0C, no priming + no offspring hypoxia stress; T0W, no priming + offspring hypoxia stress; T1C, one-generation priming + no offspring hypoxia stress; T1W, one-generation priming + offspring hypoxia stress; T2C, two-generation priming + no offspring hypoxia stress; T2W, two-generation priming + offspring hypoxia stress; T3C, three-generation priming + no offspring hypoxia stress; T3W, three-generation priming + offspring hypoxia stress. Data are means ± SE (n = 3). Different lowercase letters indicate the significant difference at *p* < 0.05 level.

**Figure 4 plants-11-01969-f004:**
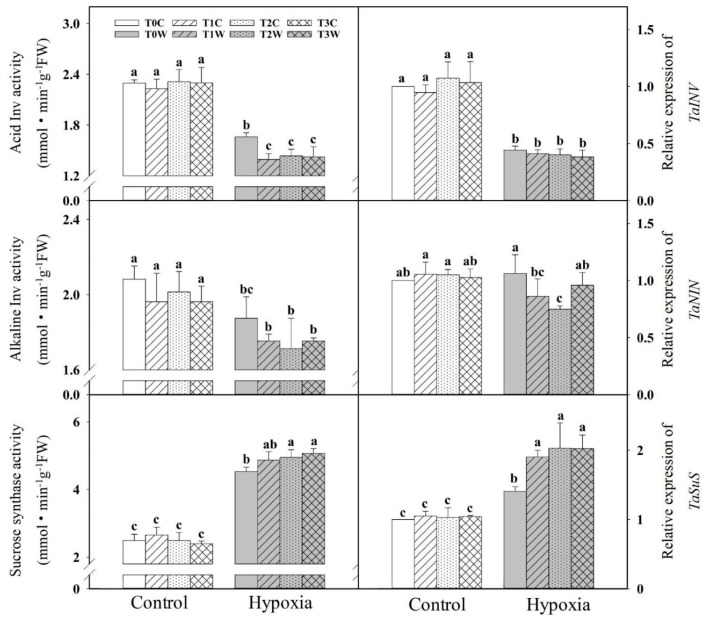
Effects of waterlogging-priming on acid invertase activity, alkaline invertase activity and sucrose synthase activity in root of offspring plants under hypoxia stress in wheat seedling. Note: *TaAINV*: acid invertase gene; *TaNIN*: alkaline/neutral invertase gene; *TaSuS*: sucrose synthase gene. T0C, no priming + no offspring hypoxia stress; T0W, no priming + offspring hypoxia stress; T1C, one-generation priming + no offspring hypoxia stress; T1W, one-generation priming + offspring hypoxia stress; T2C, two-generation priming + no offspring hypoxia stress; T2W, two-generation priming + offspring hypoxia stress; T3C, three-generation priming + no offspring hypoxia stress; T3W, three-generation priming + offspring hypoxia stress. Data are means ± SE (n = 3). Different lowercase letters indicate the significant difference at *p* < 0.05 level.

**Figure 5 plants-11-01969-f005:**
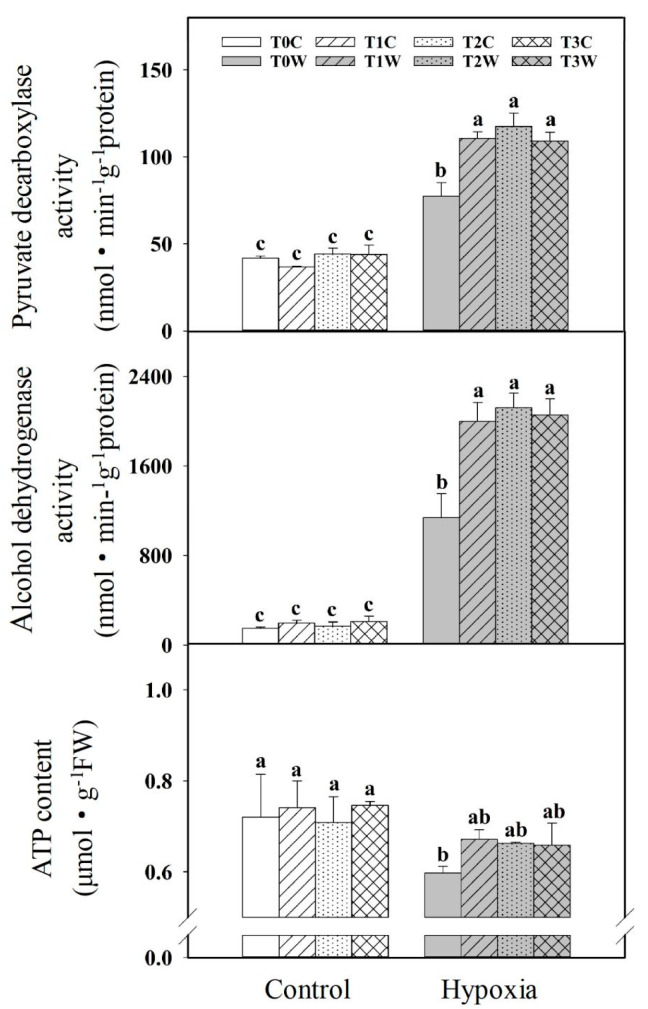
Effects of waterlogging priming on pyruvate decarboxylase activity, alcohol dehydrogenase activity and ATP content in root of offspring plants under hypoxia stress in wheat. Note: PDC: pyruvate decarboxylase; ADH: alcohol dehydrogenase; ATP: adenosine triphosphate. T0C, no priming + no offspring hypoxia stress; T0W, no priming + offspring hypoxia stress; T1C, one-generation priming + no offspring hypoxia stress; T1W, one-generation priming + offspring hypoxia stress; T2C, two-generation priming + no offspring hypoxia stress; T2W, two-generation priming + offspring hypoxia stress; T3C, three-generation priming + no offspring hypoxia stress; T3W, three-generation priming + offspring hypoxia stress. Data are means ± SE (n = 3). Different lowercase letters indicate the significant difference at *p* < 0.05 level.

**Figure 6 plants-11-01969-f006:**
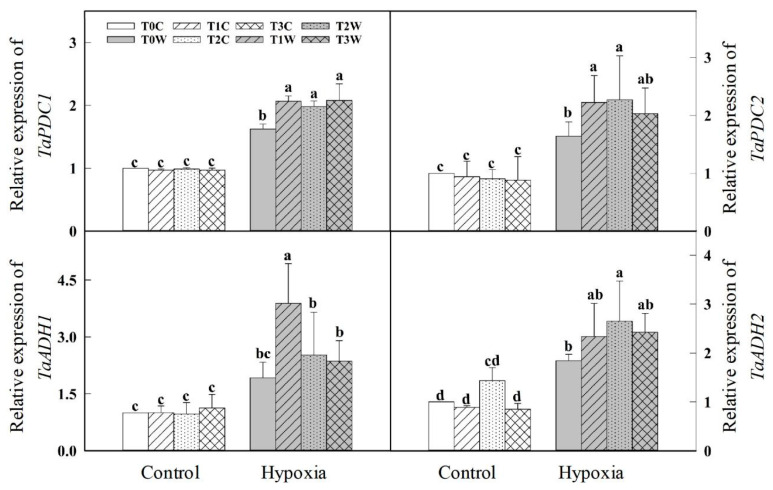
Effects of waterlogging priming on *TaPDC*, *TaADH* gene expression in root of offspring plants under hypoxia stress in wheat seedling. Note: *TaPDC1*, *TaPDC2*: pyruvate decarboxylase gene; *TaADH1*, *TaADH2*: alcohol dehydrogenase gene. T0C, no priming + no offspring hypoxia stress; T0W, no priming + offspring hypoxia stress; T1C, one-generation priming + no offspring hypoxia stress; T1W, one-generation priming + offspring hypoxia stress; T2C, two-generation priming + no offspring hypoxia stress; T2W, two-generation priming + offspring hypoxia stress; T3C, three-generation priming + no offspring hypoxia stress; T3W, three-generation priming + offspring hypoxia stress. Data are means ± SE (n = 3). Different lowercase letters indicate the significant difference at *p* < 0.05 level.

**Figure 7 plants-11-01969-f007:**
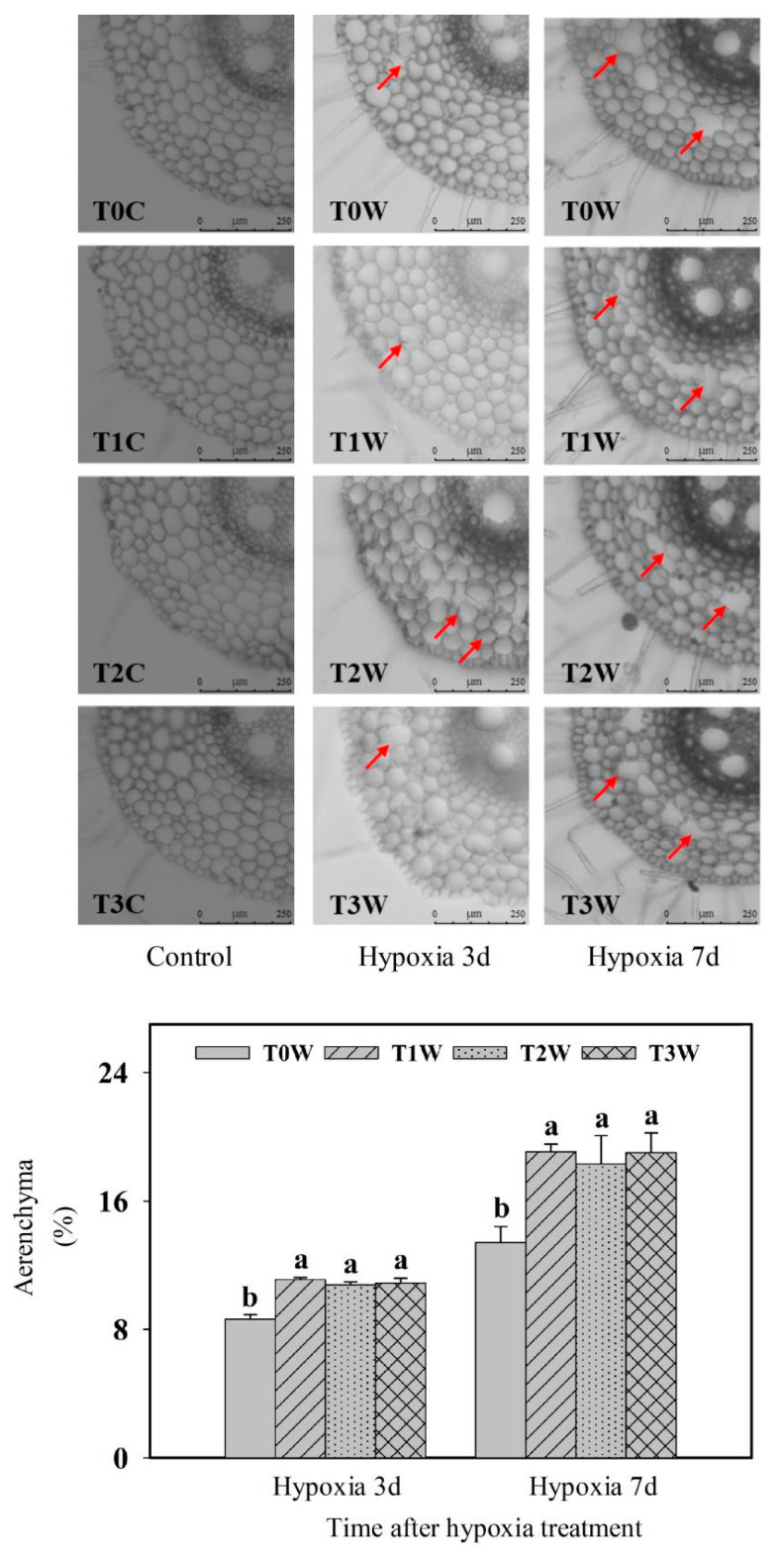
Effects of waterlogging priming on aerenchyma at the position 40mm from the root tip of offspring plants grown under hypoxia stress for 3 d and 7 d in wheat seedling. Note: The arrow indicates the position of the aerenchyma cavity. T0C, no priming + no offspring hypoxia stress; T0W, no priming + offspring hypoxia stress; T1C, one-generation priming + no offspring hypoxia stress; T1W, one-generation priming + offspring hypoxia stress; T2C, two-generation priming + no offspring hypoxia stress; T2W, two-generation priming + offspring hypoxia stress; T3C, three-generation priming + no offspring hypoxia stress; T3W, three-generation priming + offspring hypoxia stress. Data are means ± SE (n = 3). Different lowercase letters indicate the significant difference at *p* < 0.05 level.

**Figure 8 plants-11-01969-f008:**
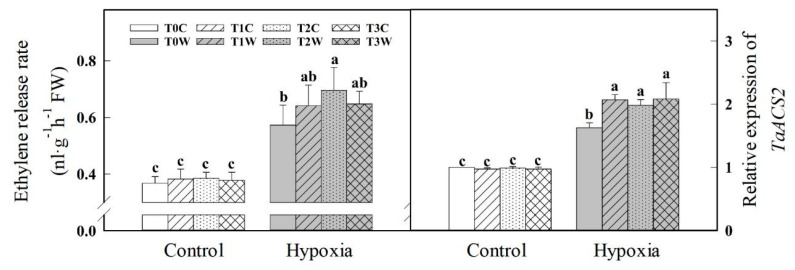
Effects of waterlogging priming on ethylene release rate and *TaACS2* gene expression in root of offspring plants under hypoxia stress in wheat seedling. Note: *TaACS2*: 1-aminocyclopropane-1-carboxylate (ACC) synthase gene. T0C, no priming + no offspring hypoxia stress; T0W, no priming + offspring hypoxia stress; T1C, one-generation priming + no offspring hypoxia stress; T1W, one-generation priming + offspring hypoxia stress; T2C, two-generation priming + no offspring hypoxia stress; T2W, two-generation priming + offspring hypoxia stress; T3C, three-generation priming + no offspring hypoxia stress; T3W, three-generation priming + offspring hypoxia stress. Data are means ± SE (n = 3). Different lowercase letters indicate the significant difference at *p* < 0.05 level.

**Figure 9 plants-11-01969-f009:**
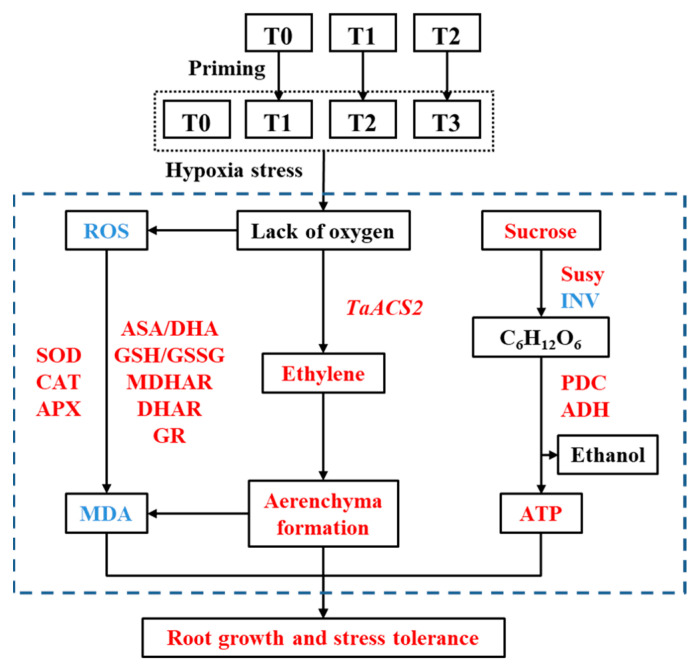
The mechanism of multigeneration waterlogging priming to improve the root resistance of offspring to hypoxia. Note: Red and blue respectively indicate the up regulation or down regulation induced by waterlogging priming under hypoxia stress.

**Table 1 plants-11-01969-t001:** Effects of waterlogging priming on function of antioxidant system in root of offspring plants under hypoxia stress in wheat seedling.

Treatment	T0C	T1C	T2C	T3C	T0W	T1W	T2W	T3W
ROS								
O_2_^−^ (µmol g^−1^Fw min^−1^)	14.10 c	14.34 c	14.52 c	14.20 c	16.64 a	15.33 b	15.13 b	15.52 b
H_2_O_2_ (mmol g^−1^Fw)	16.66 c	16.87 c	17.29 c	17.25 c	28.94 a	24.39 b	23.38 b	23.80 b
MDA (mmol g^−1^Fw)	1.60 c	1.45 c	1.49 c	1.66 c	2.85 a	2.50 ab	2.42 ab	2.24 ab
Antioxidant enzymes activities								
SOD (U g^−1^protein)	0.235 a	0.228 a	0.222 a	0.232 a	0.170 c	0.187 b	0.198 b	0.192 b
CAT (U g^−1^protein)	6.10 a	6.18 a	6.17 a	6.16 a	5.02 c	5.40 bc	5.56 b	5.53 bc
APX (U g^−1^protein)	34.39 a	34.09 a	34.37 a	33.85 ab	30.33 b	30.68 ab	33.66 ab	33.17 ab
GR (U g^−1^protein)	3.15 c	3.32 c	3.48 c	3.34 c	4.19 b	5.53 a	5.84 a	5.35 a
MDHAR (U g^−1^protein)	2.92 c	2.87 c	3.14 c	3.01 c	3.50 b	3.81 ab	3.75 ab	3.98 a
DHAR (U g^−1^protein)	3.99 c	4.11 c	3.92 c	4.23 c	5.00 b	5.58 a	5.71 a	5.65 a
Non-enzymatic antioxidants								
AsA (mg g^−1^Fw)	0.13 c	0.14 c	0.13 c	0.11 c	0.27 b	0.41 a	0.41 a	0.40 a
DHA (mg g^−1^Fw)	0.33 d	0.34 d	0.34 d	0.34 d	0.56 c	0.67 b	0.69 b	0.73 a
AsA + DHA	0.46 c	0.48 c	0.47 c	0.45 c	0.83 b	1.08 a	1.10 a	1.13 a
AsA/DHA	0.41 cd	0.42 cd	0.38 d	0.33 d	0.49 bc	0.62 a	0.59 a	0.54 ab
GSH (mg g^−1^Fw)	11.06 c	11.45 c	11.32 c	11.43 c	12.41 b	14.84 a	14.93 a	15.10 a
GSSG (mg g^−1^Fw)	2.72 a	2.60 a	2.66 a	2.70 a	2.10 b	1.77 b	1.81 b	1.87 b
GSH + GSSG	13.78 b	14.05 b	13.98 b	14.13 b	14.51 b	16.61 a	16.74 a	16.97 a
GSH/GSSG	4.14 c	4.49 c	4.38 c	4.24 c	5.92 b	8.39 a	8.26 a	8.13 a

Note: T0C, no priming + no offspring hypoxia stress; T0W, no priming + offspring hypoxia stress; T1C, one-generation priming + no offspring hypoxia stress; T1W, one-generation priming + offspring hypoxia stress; T2C, two-generation priming + no offspring hypoxia stress; T2W, two-generation priming + offspring hypoxia stress; T3C, three-generation priming + no offspring hypoxia stress; T3W, three-generation priming + offspring hypoxia stress. Data are means ± SE (n = 3). Different lowercase letters indicate the significant difference at *p* < 0.05 level.

## Data Availability

The data presented in this study are available within the article.

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
