# Peer review of "Waterlogging Priming Enhances Hypoxia Stress Tolerance of Wheat Offspring Plants by Regulating Root Phenotypic and Physiological Adaption"

_plants, 2022, doi:10.3390/plants11151969_

Round 1

Reviewer 1 Report

Waterlogging priming enhances hypoxia stress tolerance of wheat offspring plants by regulating root phenotypic and physiological adaption 

This article concern waterlogging priming and testing of the first, second and third generation of wheat under hypoxia stress. The authors of the article found that earlier priming enhances tolerance to hypoxia by improving the photosynthetic and the antioxidant system. The work was carried out comprehensively at the physiological, biochemical, molecular and anatomical levels. 

The results of the work are interesting, the article is quite well written, although the authors did not avoid minor mistakes. 

Introduction line 64-65- stylistically incorrect sentence. 

Minor mistake, please put a space before each quotation- this applies to the entire article. 

Materials and methods- subsection antioxidant activities - please unify - dismutase and catalase activity, please describe in more detail, like the rest of the enzymes or the non-enzymatic antioxidant system. 

Discussion - it is too long and at the same time it seems to duplicate description of the results at times. I miss a deeper discussion with other authors who describe the plant's response to hypoxic stress. 

Literature - some journals names are written with a capital letter, others with a lowercase letter, some with an abbreviation and the other not. Please bring it in line with the current rules of the Plants journal. 

Good luck!

Reviewer 2 Report

The manuscript describes the comparison of generation-dependent waterlogging priming of wheat. The whole manuscript is well-structured and contains valuable results. Abstract is good written but in L18 there is no verb so please check it. Keywords are also well selected. Introduction is also good for demonstrate the background information to understand this work.  However, there are some grammatical mistakes, so I recommend this work to be checked by a native English speaker. In the results, I cannot understand why authors wrote “seeds” when they investigate the whole plant leaves and roots? I advise to rewrite the titles of results. In case of Table 1, I highly recommend to calculate the total ascorbate and total glutathione and also the redox ratio of them, red Asc/total Asc and red GSH/total GSH. These parameters also show us that what could happen with these valuable antioxidants .Is it only one gene coding the invertases in wheat, If not, please describe why authors selected this gene. In the L214, this sentence is not good. Authors did not investigate different root positions, only one was selected for analysis, the 40 mm from root tip. I suggest that this sentence should be rewritten. The Figure 7 legend is very short, please describe precisely the conditions of photo making, scale bar is also missing. In case of ethylene the same problem, only one gene could be involved in ethylene synthesis in wheat? please provide information about selection. In Mateials and methods, there are some missing information about the companies, eg. L399519,522, 532. In 4.4 should be antioxidant enzyme activities and antioxidants level. In L468, this sentence is not correct, please check it. OD numbers should be smaller in lower index. Also, the Systat Software company place is missing in L549.  I do not understand why Figure 9 is needed as in current form it does not show any multi-generation features, please remodel it precisely. References are written in a different manner and not as MDPI wants, so please write it correctly.

Reviewer 3 Report

This work presents some novelty and includes analysis of a range of parameters highly informative for waterlogging stress. However, the English needs substantial improvement, especially grammar and tenses; at places the English is not clearly understandable. English editing is indispensable.

Priming and trans generational effects  - needs more justification in the Introduction. Only two references by the same affiliation and part of the authors, on drought and high temperature stresses  – it seems that the research is broadening but not deepening. The authors should mention in the introduction the epigenetic events in stress memory, with appropriate references.  

MMs – Growth conditions and priming treatment are not sufficiently explained. This section needs substantial improvement.

 Line 380- “After 379 priming, the pots were re-watered similarly to the level of control pots” – why re-watering after waterlogging?

 Was priming repeated for 1,2,3,generations? It seems to be repeated priming treatment (in S1fig) but this becomes clear in the main text only in the discussion.

Priming after anthesis T0 soil cultures, but stress after priming - at seedling stage 4th leaf before anthesis , in hydroponics – what was the reason for choosing different type of experiment?

Line 412 "The Oâ‚‚.- release rate was assayed by hydroxylamine method[43]. "– this method is for SOD activity, please explain how you used it for detection of superoxide level. …” the color was compared under 530 nm ultraviolet and visible light source” – unclear, 530 nm is visible light

Line 418- enzyme extraction for MDA?

4.9. RNA Extraction and Quantitative Real-Time PCR – which transcripts have been analysed?  Not clear in MMs and from the supplementary material. Only in Results it becomes clear that these genes are TaPDC1, TaPDC2: Pyruvate decarboxylase gene; TaADH1, TaADH2: Alcohol dehydrogenase gene; TaACS2: 1-aminocyclopropane- 1-carboxylate (ACC) synthase gene.

Description of the methods for enzyme activities are like an instruction in lab protocol – weight sample, grind it, … MMs section needs re-writing and English editing.

Results

Fig 1 legend – why “of offspring plants leaf” – there is root and shoot FW and DW? “Lowercase letters” or letters above columns? For growth parameters 3 replicas are too low number.

Line 125 – SOD activities “which were still lower than…” – unclear, actually the decrease is lower, activities are higher than T0W.

Results presented in Table 1 will be much more easily perceived if put in 2-3 figures.

Why sugars were measured both in roots and leaves while enzyme activities for sugar metabolism – only in roots? Oxidative stress markers were also estimated only in roots.

Discussion

Line 271 – “Waterlogging stress led to less oxygen and could lead to oxidative damage to plants” – how the authors could explain it?

Line  282 – “Waterlogging treatment could increase antioxidant enzyme activity” – in Results SOD and CAT activities are diminished contrary to the enzymes of ascorbate-glutatione cycle – any explanation?

Line 290 – inhabit?

Round 2

Reviewer 3 Report

The English language of the manuscript still remain full of gramatical errors and needs substantial editing

How the authors could explain the lack of cumulative effect of the successive primings on the improvement of plant performance under waterlogging?
